# Handy divisions: Hand-specific specialization of prehensile control in bimanual tasks

**Anvesh Naik**[�][¤], **Satyajit Ambike**[iD][�they][*]

Department of Health and Kinesiology, Purdue University, West Lafayette, Indiana, United States of America,

they These authors contributed equally to this work.
¤ Mortimer B. Zuckerman Mind Brain Behavior Institute, Columbia University, New York, USA
* sambike@purdue.edu

## Abstract

When hammering a nail, why do right-handers wield the hammer in the right hand? The complementary dominance theory suggests a somewhat surprising answer. The two hands are specialized for different types of tasks: the dominant for manipulating objects, and the non-dominant for stabilizing objects. Right-handers wield the moving object with their right hand to leverage the skills of both hands. Functional specialization in hand use is often illustrated using examples of object manipulation. However, the complementary dominance theory is supported by wrist kinematics rather than object manipulation data. Therefore, our goal was to determine whether this theory extends to object manipulation. We hypothesized that hand-specific differences will be evident in the kinematics of hand-held objects and in the control of grip forces in right-handed individuals. Right-handed participants held two instrumented objects that were coupled by a spring. They moved one object while stabilizing the other object in various bimanual tasks. They performed motions of varying difficulty by tracking predictable or unpredictable targets. The two hands switched roles (stabilization vs movement) in various experimental blocks. The changing spring length perturbed both objects. We quantified the movement performance by measuring the objects' positions, and grip force control by measuring grip-load coupling in the moving hand and mean grip force in the stabilizing hand. The right hand produced more accurate object movement, along with stronger grip-load coupling, indicating superior predictive control of the right hand. In contrast, the left hand stabilized the object better and exerted a higher grip force, indicating superior impedance control of the left hand. Task difficulty had a weak effect on grip-load coupling during object movement and no effect on mean grip force during object stabilization. These behavioral results demonstrate that complementary dominance extends to object manipulation, though the weak effect of task difficulty on grip characteristics warrants further investigation. Neurophysiological investigations can now examine the hemisphere-specific neural mechanisms underlying these behavioral differences.

## Introduction

Approximately 90% of humans favor their right hand for manual tasks [1]. This asymmetry in hand use has intrigued scientists since the time of Woodworth [2], and has led to the

**Data availability statement:** URL: https://purr. purdue.edu/publications/4436/3 doi: 10.4231/ Q7SB-2650

**Funding:** Anvesh Naik was funded by The Ross Fellowship awarded by Purdue University. The funders had no role in study design, data collection and analysis, decision to publish, or preparation of the manuscript.

**Competing interests:** The authors have declared that no competing interests exist.

development of two major theories explaining the motor control behind handedness. The global hemispheric dominance theory posits that the dominant (right) hand is specialized for all aspects of movement control compared to the non-dominant (left) hand; and because each hand is controlled mostly by the contralateral hemisphere, this theory suggests that the left hemisphere assumes a dominant role in right-handed individuals for controlling both hands [3–6]. In contrast, the complementary dominance theory (formerly known as the dynamic dominance theory) proposes that each hemisphere specializes in different aspects of movement control [7–10]. The left hemisphere specializes in predictive control; it anticipates changes in the environment and arm dynamics and generates coordinated movements with the right arm better than the right hemisphere can produce coordinated movements with the left arm. In contrast, the right hemisphere specializes in impedance control (Table 1); it stabilizes the left arm against perturbations better than the left hemisphere can stabilize the right arm. This view was based on seminal behavioral data gathered about four decades ago [11,12], which Sainburg and colleagues subsequently formalized into the complementary dominance theory based on wrist kinematics during reaching movements in healthy individuals [8,9,13] and unilateral stroke survivors [14,15].

Here, our primary goal was to move beyond wrist kinematics to determine whether the complementary dominance theory explains the manipulation of hand-held objects in right-handed individuals. More importantly, we were interested in determining whether the control of the forces exerted by the digits on the object was also described by this theory. It is plausible that hemispheric specialization for controlling arm movements extends to the control of digit force, but this is not inevitable for two reasons. First, non-overlapping neural substrates are involved in the control of arm movement and digit forces [16–21]. Second, behavioral evidence indicates that the control of arm movements and grip forces (Table 1) is partly dissociated [22]. For example, humans adjust the grip force on a static object *before* they start moving the object, indicating that grip forces can change even when the object's motion, and therefore arm motion, is the same [23]. Therefore, behavioral evidence for complementary dominance in object manipulation is essential for establishing the domain of applicability of this theory.

Our strategy to look for this evidence was to inspect the grip force-load force coupling (Table 1; henceforth grip-load coupling for brevity) and the magnitude of grip force. These variables are critical for manipulating objects stably, and they reflect the predictive and

**Table 1. Definitions and descriptions of key terms.**

| Term | Definition/Description |
|---|---|
| Grip force | Equal and opposite contact forces exerted on a grasped object by the thumb and fingers. These forces are orthogonal to the digit-object interface. |
| Load force | The vector sum of all forces on a grasped object that are not exerted by the hand. These forces are: object weight, all forces exerted by the environment, and inertial force that arises from the object's motion. |
| Grip-load coupling | A feature of human prehension control. It is a feedforward process whereby humans modulate grip force in synchrony with changes in load force, for example, while volitionally moving or oscillating a grasped object. |
| Mechanical Impedance | Impedance is the dynamic relation between force and position (and its derivatives). It describes how a system resists motion when subjected to an external force, and how that resistance arises from the system's configuration (position-dependent stiffness), velocity (damping) and acceleration (inertia). |
| Impedance control | A strategy for controlling movement where the nervous system regulates hand (endpoint) impedance by modulating the mechanical properties (stiffness, damping, and inertia) of the arm (linked chain of joints) in the musculoskeletal system. This regulation emerges from both feedforward setting of neuromuscular states based on expected task demands and feedback corrections based on sensed interactions. Rather than controlling either force or position independently, the system controls how the hand responds to interaction forces with the environment. |
| Impedance control at the object level | An extension of impedance control where the object becomes the endpoint of the chain. The nervous system regulates object impedance by combining control of hand impedance with that of the grip forces that transmit the hand's impedance to the object. |

impedance control aspects of prehension. The predictive aspect of the control is demonstrated by humans coupling the grip force with the predicted loads on the grasped object [24–29]. The impedance control of the object (or object-level impedance; Table 1) would be regulated by the nervous system through the control of the grip forces that transmit the hand's impedance to the object. Humans exert grip force exceeding the minimum required to ensure a stable slip-free grasp [26,28,29]. A higher grip force also stiffens the wrist through muscle co-contraction, further improving the object's positional stability [29–32]. Thus, if complementary dominance extended to grip force control, then the grip-load coupling would be stronger in the dominant (right) hand when moving an object and the grip force magnitude would be higher in non-dominant (left) hand when stabilizing an object.

Additionally, our secondary goal was to quantify how task difficulty interacts with complementary dominance during bimanual prehensile tasks. Recent studies on unimanual prehension reported that grip force characteristics are altered by task difficulty. Grip-load coupling is stronger when individuals track unpredictable compared to regular and predictable paths with a handheld object [33,34]. Furthermore, the grip force increases when external loads act on the object, but it increases more when the loads are variable [35]. We argue below that complementary dominance may not manifest for simple tasks; therefore, we systematically studied how complementary dominance is influenced by task difficulty.

Previous studies on grip force control lacked evidence for complementary dominance, possibly due to task design issues. Jaric and colleagues used isometric bimanual force production tasks with fixed handles [36–40]. They found similar grip-load coupling in both hands, and lower grip force in the left hand [39], results that are incompatible with the predictions of complementary dominance theory [8,41]. The issue here may have been that the static tasks in this study favored the left hand and masked effects that could emerge while moving objects.

Other studies involved object movement, but still failed to observe between-hand differences. Participants exhibited similar grip-load coupling and grip force in both hands while moving two objects with different weights [42,43] or holding one object steady while simultaneously moving another object [44,45]. These results may stem from ceiling effects; the tasks may have been too simple, and both hands could perform the tasks equally well without significantly changing grip forces [46].

These studies revealed two crucial considerations for studying complementary dominance in prehension. First, tasks should include both movement and stabilization to leverage the specialization of each hand. Second, the task must be sufficiently challenging to reveal hand-specific differences in grip force control. Based on this thinking, we designed a bimanual task that approximated bread slicing: one hand stabilized an object against the disturbances arising from the movements of the other hand [13]. Participants manipulated two objects, one per hand, connected by a spring. They tracked a visual target by moving one object while holding the other object stationary. The target profiles were predictable or unpredictable in different conditions, which altered task difficulty. This task challenged the motor system by requiring distinct actions from each hand while accounting for the spring-induced disturbances.

Our working hypothesis was that complementary dominance would extend to object kinematics and grip force control. The right hand will show superior object movement accuracy because the left hemisphere specializes in predictive control (hypothesis H1). In contrast, the left hand will show superior object stabilization because the right hemisphere specializes in impedance control (hypothesis H2). These hypotheses extend the classical observations on wrist kinematics [8,13] to the control of hand-held objects. Furthermore, the grip-load coupling strength – the predictive aspect of prehension – will be higher for the right hand when moving the object (hypothesis H3). The mean grip force – the impedance aspect of

prehension – will be higher for the left hand while stabilizing the object (hypothesis H4). Finally, these grip force characteristics will show an interaction between hand and task difficulty. A more difficult task will strengthen the grip-load coupling in both hands while moving objects, but the increase will be greater in the right hand than the left (hypothesis H5). In contrast, a more difficult task will increase the mean grip force in both hands while stabilizing objects, but the increase will be greater in the left hand than the right (hypothesis H6).

## Materials and methods

### Participants

Twenty-four, young individuals (11 females; age = 24.7 ± 4.6 years; weight = 73.2 ± 14.6 kg; height = 168.7 ± 10.6 cm) volunteered to participate in the study. Handedness was assessed by the Edinburgh Inventory [47]. This inventory provides a laterality quotient (LQ) ranging from -100 (strong left-handedness) to +100 (strong right-handedness). All participants were right-hand dominant (LQ = 88.5 ± 9.5). Only healthy individuals with no sensorimotor impairments or trauma to upper limbs were included in the study.

### Ethics statement

All participants provided written informed consent in accordance with the procedures approved by the Institutional Review Board of Purdue University. IRB approval was obtained on February 6, 2023 (Number: IRB-2023–48). The recruitment period for this study was from February 27, 2023, to May 12, 2023.

### Instrumentation

Participants held two instrumented objects in a pinch grasp with the tips of index finger and thumb of each hand (Fig 1). Six-component force transducers (Nano 17-E, ATI Industrial Automation, Garner, NC) mounted on the objects measured the force of each digit. The distance between the grasping surfaces of the two force transducers on an object was 6.5 cm. Sandpaper (100C medium grit) was glued to the surface of transducers to increase the coefficient of friction between the transducer and the digits. At the start of each experimental session, force transducers were zeroed with objects resting vertically on the table, and no digits contacting the sensors. A tension spring (stiffness 19.5 N/m, and resting length 8 cm) was attached between the two objects (Fig 1). When the spring was at its resting length, the horizontal distance between the centers of the sensors on the two objects was 13 cm. The total weight of the apparatus was 2 N. Four reflective markers were attached to each object (Fig 1) and a four-camera motion capture system (Vicon Vero VE22-S, Oxford, UK) was used to track the position of both objects. The motion capture system was calibrated for each participant to obtain a tracking error less than 1 mm inside the capture volume. The MotionMonitor software (Innovative Sports Training Inc.) was used to synchronize and collect output signals from the force transducers (sampled at 1000 Hz) and the motion capture system (sampled at 200 Hz).

### Experimental setup and procedure

Before the start of an experimental session, participants cleaned the digit tips of both hands using alcohol wipes to normalize the skin condition. Participants sat upright on a piano bench facing a computer screen placed on a table (Fig 2). A reference point was marked on the table as the origin for the world co-ordinate system. The positive X axis pointed to the participant's right, the positive Y axis pointed along the anterior direction, and positive Z axis pointed vertically upward.

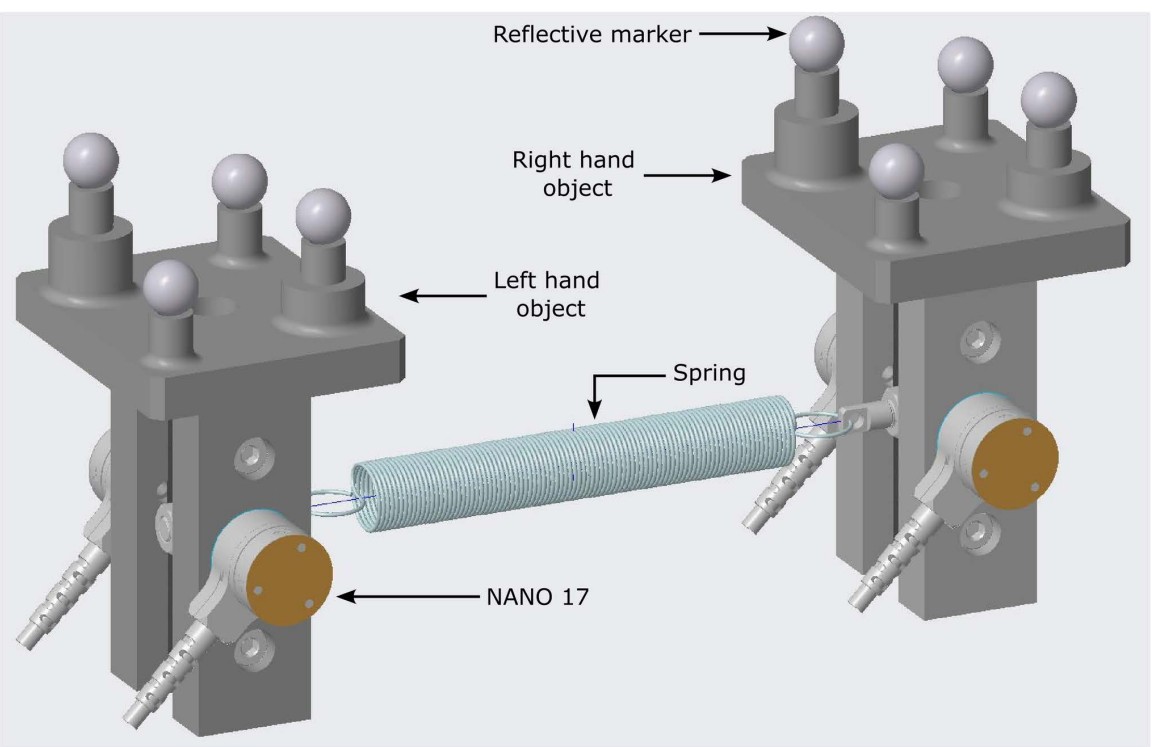

**Fig 1. Illustration of the instrumented objects.** Nano 17 force sensors recorded digit forces. A motion capture system tracked the positions of four reflective markers fixed to each object.

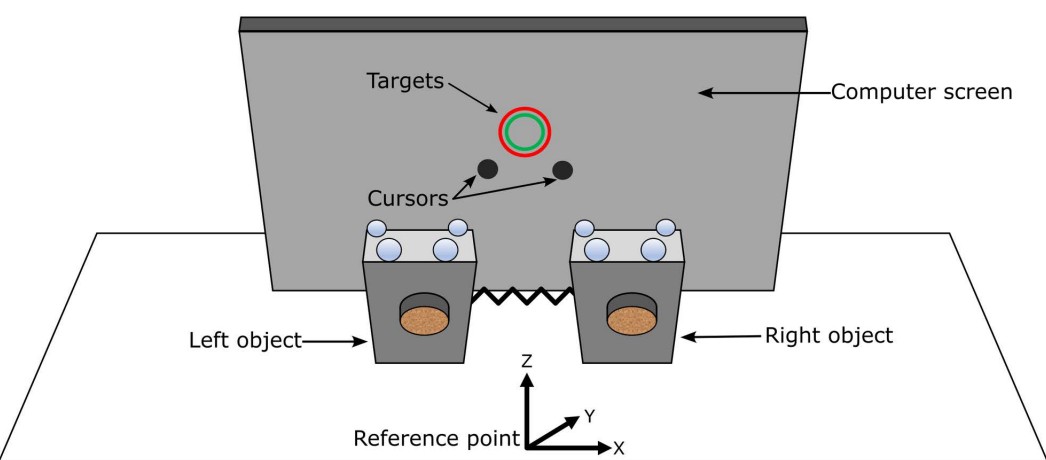

**Fig 2. Experimental setup.** Participants held the objects in pinch grasps and positioned them above the reference point. Targets and cursors were displayed on the computer screen placed on the table.

The participants grasped both objects by placing the index finger and the thumb of each hand on the transducers. They lifted the objects and held them vertical and stationary at a comfortable height above the reference point (Figs 2 and 3A). The computer screen displayed two hollow circular *targets* and two solid circular *cursors* (Fig 3A). The cursors indicated the real-time positions of the objects in the frontal X-Z plane. The centroid of the four markers

on the object was defined as the position of that object. The cursors were flipped on the screen so that the cursor on the left represented the right-hand object and vice versa. Although the initial positions of cursors were flipped, their motions on the screen were veridical. That is, the rightward motion of any of the objects moved the corresponding cursor to the right on the screen, and so on for the other horizontal and vertical directions (Fig 3A). Participants moved both objects vertically and apart from one another until the two cursors overlapped at the center of the targets (Fig 3B). In this position, the spring was stretched by approximately 3 cm which generated a baseline horizontal spring force of approximately 0.6 N on each object.

In each trial, the green target moved vertically on the screen, whereas the red target remained stationary (Fig 3C). Participants were instructed to move one object vertically so that the corresponding cursor tracked the green target as accurately as possible, while simultaneously stabilizing the other cursor inside the stationary red target (Fig 3C). The red target was bigger in size than the green target to prevent both targets from overlapping. At the end of each trial, targets and cursors vanished and participants replaced both the objects on the table and released the grasp.

There were four blocked experimental conditions (2 *hand* × 2 *task difficulty*), with repeated trials in each block. In two blocks, the right hand moved one object, and the left hand stabilized the other object. In the other two blocks, the role of each hand was switched. The task difficulty was modulated by changing the moving target's trajectory. In different blocks, the movement profile was a regular and predictable triangular wave (amplitude 0.24 m, frequency 0.5 Hz). In the other two blocks, the profile was an irregular and unpredictable fractional Brownian motion (fBm; Hurst exponent 0.25). The same fBm sequence was used in all trials for all participants. The two profiles and the tracking performance by each hand by one representative participant are shown in Fig 4. The order of each *hand* × *task difficulty* block was counterbalanced across participants using the Latin square design.

There were 10 trials in each block and 40 trials in total during the experimental session. Each trial was 15 s long. In the first 4 seconds, the targets remained in the initial position and participants stabilized both cursors inside the targets (Fig 3B). The target movement began at the 4th second and lasted until the end of the trial. At the start of each block, participants were informed which hand to move and which hand to stabilize. Participants were given adequate practice before each block till the experimenter was satisfied with the task performance. After each trial, a score reflecting the participant's performance was displayed on the screen (Fig 3D). The score was the sum of the mean squared error between the center of each cursor to the center of the corresponding target. Participants were instructed to keep this score as low as possible.

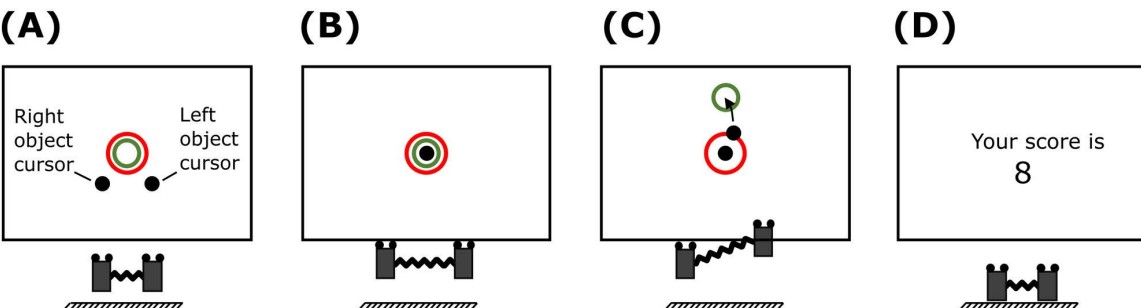

**Fig 3. Targets and cursors feedback on the computer screen.** (A) position of cursors and objects at the start of a trial, (B) objects positioned in the frontal plane such that the spring is stretched to bring both cursors within the targets, (C) target tracking with right object while left object is stabilized, and (D) score displayed at the end of the trial.

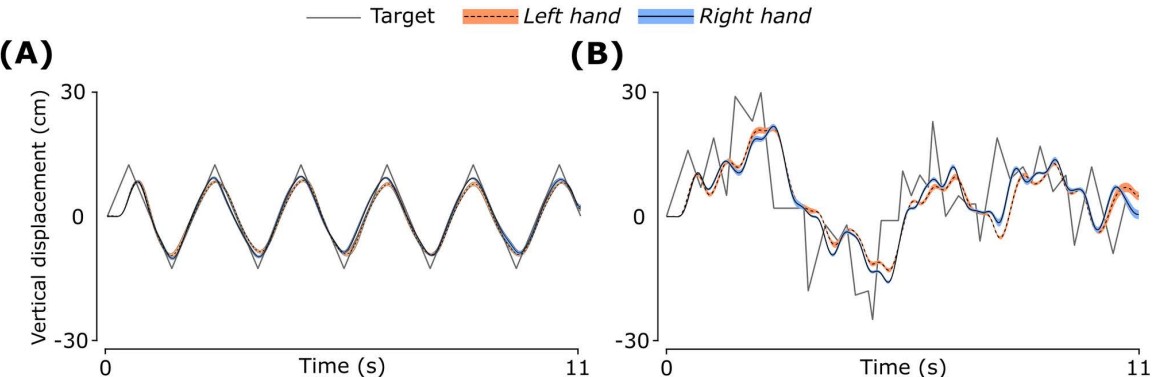

**Fig 4. Target profiles and tracking performance by each hand.** (A) Regular (triangular) movement and (B) Irregular (fBm) movement. Data shown are across trial mean and standard error for a representative participant.

Participants rested for 15 s between trials, and for 60 s between blocks. Additional rest periods were provided when participants felt fatigue in their arms or digits, and they were encouraged to ask for additional rest whenever required. The entire protocol lasted for about an hour. None of the participants requested additional rest, and none of the participants reported fatigue during the protocol.

## Data analysis

Custom MATLAB programs were written for data analyses (R2023a, The MathWorks Inc). To match the sampling frequencies of the kinematics and the digit forces, the force data was downsampled to 200 Hz. The down-sampled finger forces and kinematic data were low-pass filtered at a cutoff frequency of 4 Hz using a fourth-order, zero-lag Butterworth filter [34]. The data within an *analysis window* between 4 s and 15 s was used for further analysis.

The grip force for each hand was computed as the sum of the index finger and thumb normal forces of that hand [48]. The load force on each object was computed as the vector sum of tangential forces along the sensor-digit interface on both force transducers on an object. The inertial force vector was computed for the moving object as the object's mass times its acceleration in the X-Z plane. The spring force on the moving object was then obtained by subtracting the load and weight vectors from the inertial force vector. Since the spring connected the two objects, the same spring force acted on the stabilized object as well. Profiles of the grip and load forces from a representative participant are shown in Fig 5.

**Basic performance measures.** To quantify *task difficulty*, we computed performance score for each trial. To test whether participants stretched the spring before movement initiation, the spring length at the start of the analysis window (4 s after trial initiation) was quantified. To determine whether the orientation of both objects was vertical, we quantified object tilt (°) as the angle between the force transducer's plane and the X-Y plane since only this tilt influenced the load force. Note that we did not provide biofeedback on object tilt. Therefore, we did not use object tilt as a measure of task performance. We quantified the tilt to confirm that the deviation in object's orientation with respect to vertical was low, and thereby had small impact on the computation of load force. Finally, we computed the mean and variance of the load force on both objects within the analysis window to quantify the effect of *task difficulty* on characteristics of load force.

**Task performance measures.** To assess tracking performance of the moving hand, the root mean squared error (RMSE) between the centers of the moving target and the cursor

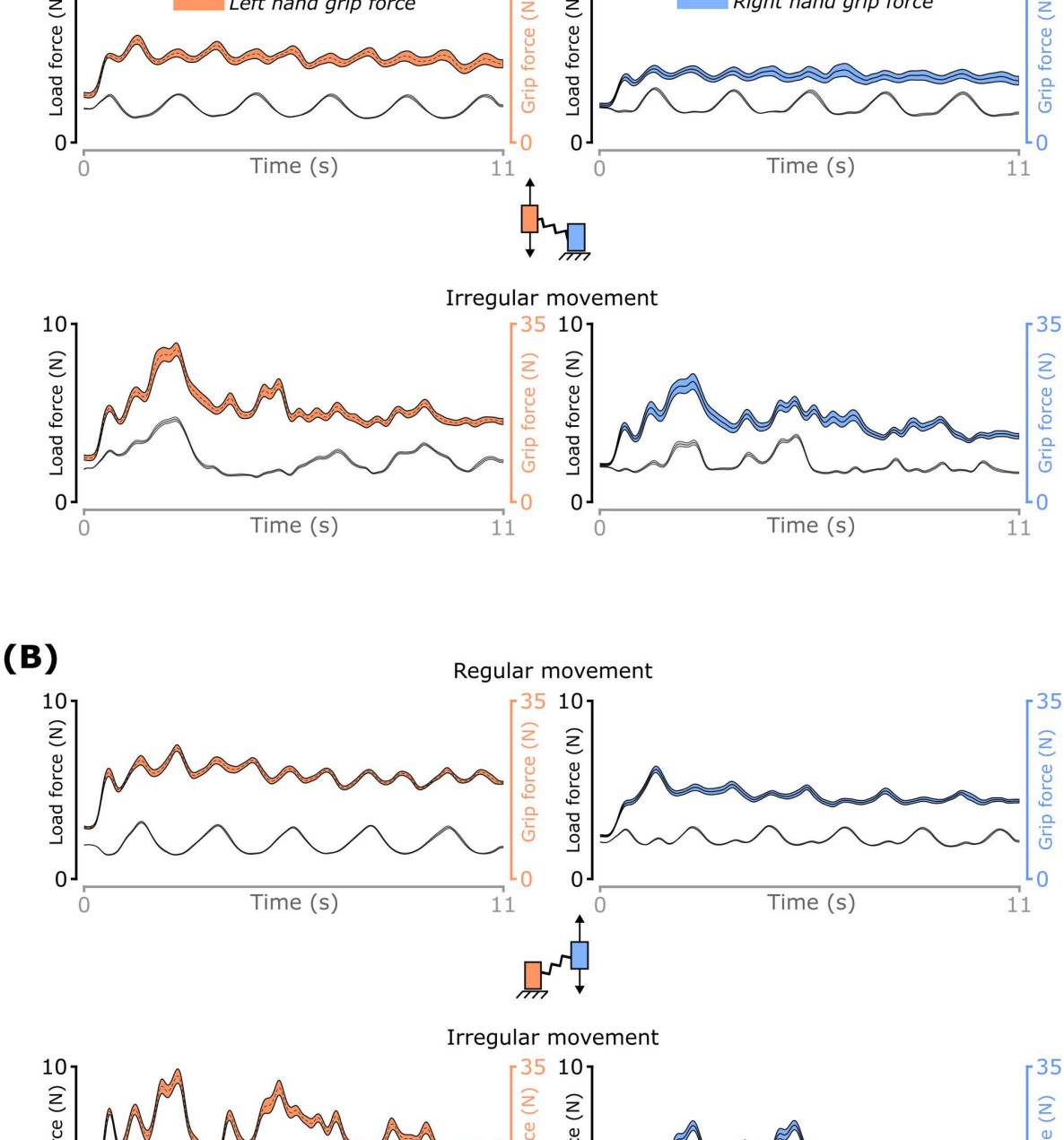

**Fig 5. Grip and load forces for both hands.** (A) moving left and static right hand during regular and irregular movements with the left hand, (B) moving right and static left hand during regular and irregular movements with the right hand. Data shown are across trial mean and standard error for a representative participant. The plots use different Y axes for grip and load forces.

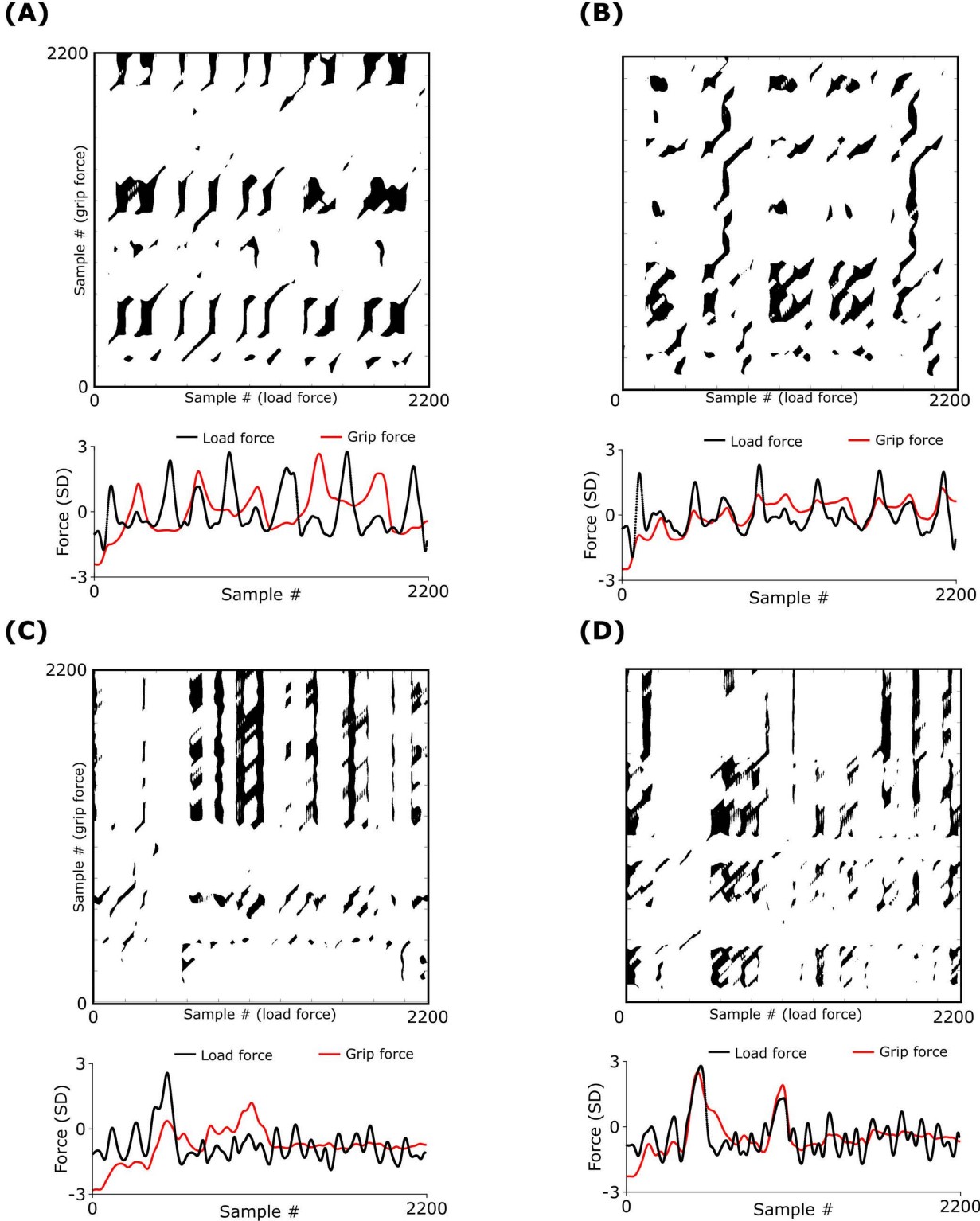

**Fig 6. Cross-recurrence plots for a representative subject.** (A) Regular movement with left hand, (B) regular movement with right hand, (C) irregular movement with left hand, and (D) irregular movement with right hand. Normalized grip and load forces are shown below each cross recurrence plot. Note that the grip-load coupling analysis was done during object movement by each hand. Therefore, the plots shown are for each hand during their respective movement tasks.

were computed separately along the vertical and mediolateral directions. A lower RMSE along the vertical indicates better tracking performance and lower RMSE along the mediolateral direction implies straighter movements.

To assess the performance of the stabilizing hand at maintaining the object's position, the standard deviation of the object's position was computed in the frontal plane. A difference in the standard deviation across conditions could arise from difference in the external load. Therefore, the standard deviation was normalized by the spring force over the trial to obtain the compliance of the stabilized object [13]. The spring force was computed as the mean of the magnitudes of the spring force vectors within a trial. Technically, the external load on the object is the sum of the spring force and the object weight. As the object weight was constant, it was not included in the compliance calculation. Lower standard deviation and compliance imply better stabilization.

**Measures of grip force control.** The grip-load coupling was quantified for the moving hand using linear cross-correlational analysis and a non-linear technique called cross-recurrence quantification analysis (CRQA). The mean grip force was computed for the stabilizing hand by averaging the grip force over the analysis window for each trial.

**Cross-correlational analysis.** Linear cross-correlational analysis was performed to determine the correlation at zero lag [49]. We computed correlation at zero lag and not maximum correlation because approximately 80% trials (772 out of 960 trials) had zero lag between grip and load forces.

**Cross-recurrence quantification analysis.** The cross-recurrence quantification analysis was performed to quantify subtle temporal variations in grip-load coupling that cross-correlational analysis fails to capture [33,34,50]. A detailed explanation of this analysis is provided in S1 Appendix. We constructed the cross recurrence plot for grip and load force time series for every trial and consistently observed vertical line structures (Fig 6). This suggest that the state of one process does not change or changes slowly relative to the state of the other process, i.e., one time series gets trapped at a location while the other deviates from that location [51]. This implies the coupling between the two processes is intermittent [33,52], and the degree of intermittency can be used to quantify the strength of the coupling. In particular, the strength of the coupling is inversely proportional to the intermittency, which, in turn, can be quantified using characteristic length of the vertical lines. This characteristic length was computed using trapping time, which is the average length of all the vertical lines in a plot [52,53].

## Statistics

The data are presented as mean ± standard errors (SE) in the Results section, unless mentioned otherwise. All outcome measures were averaged across 10 trials for each *hand × task difficulty* block. Normality and constant variance requirements were checked visually by plotting studentized residual plots for each averaged outcome measure. The Box-Cox transformation was applied when the requirements were violated. The correlation coefficient values were z-transformed to meet requirements of normality. However, non-transformed data are presented in the Results section. Then, separate linear mixed-effects (LME) models [54] were fit to each basic performance measures, task performance measures, and measures of grip force control. The two experimental factors (*hand × task difficulty*) and their two-way interactions were included as fixed effects and *participant* was included as a random effect. Tukey-Kramer test was used to perform post-hoc pairwise comparisons when significant interaction effects were observed. Effect sizes were quantified by computing Cohen's *d*. All statistics were performed using the SAS statistical software (version 9.4; SAS Institute, Cary, NC), with an α-level of 0.05.

## Results

### Basic performance measures

Participant score (Table 2) was not affected by *hand* ($F_{(1,23)} = 0.7$; $p = 0.4$) but was affected by *task difficulty* ($F_{(1,23)} = 4182$; $p < 0.01$). Post-hoc analysis revealed that the score was lower during regular movement compared to irregular movement (Cohen's $d = 13$).

The stretched spring length before movement initiation was not affected by *hand* ($F_{(1,23)} = 1.7$; $p = 0.2$) or *task difficulty* ($F_{(1,23)} = 0.3$; $p = 0.60$), when both objects were stabilized with the respective cursors inside the green target (see Fig 5B in Methods). The center-to-center distance between the objects before movement initiation averaged across *hand* and *task difficulty* and participant was 16 cm ± 0.2 cm, indicating that participants stretched the spring initially by 3 cm as required.

The mean tilt (°) of the moving object (Fig 7A) was affected by *hand* ($F_{(1,23)} = 17.0$; $p < 0.01$), but not by *task difficulty* ($F_{(1,23)} = 0.21$; $p = 0.65$). Post-hoc analysis revealed that the mean tilt was lower for the right object compared to the left object (Cohen's $d = 0.8$). The mean tilt of the stabilized object (Fig 7B) was also affected by *hand* ($F_{(1,23)} = 20.5$; $p < 0.01$) but not by *task difficulty* ($F_{(1,23)} = 0.39$; $p = 0.53$). Post-hoc analysis revealed that the mean tilt was lower for the right object compared to the left object (Cohen's $d = 0.9$).

The mean load force on the moving object (Fig 8A) was affected by *hand* ($F_{(1,23)} = 4.9$; $p = 0.04$) and *task difficulty* ($F_{(1,23)} = 74.0$; $p < 0.01$). Post-hoc analysis revealed that the mean load force was higher for the right object compared to the left (Cohen's $d = 0.4$) and lower during regular movement compared to irregular movement (Cohen's $d = 1.7$).

**Table 2. Summary of participant performance scores for each *hand* × *task difficulty* block. The values computed are across participant average ± S.E. within each block.**

|  | Left – regular | Right – regular | Left – irregular | Right – irregular |
|---|---|---|---|---|
| Mean ± S.E. (cm²) | 8.43 ± 0.5 | 7.81 ± 0.4 | 77.6 ± 1.9 | 78.7 ± 2.3 |

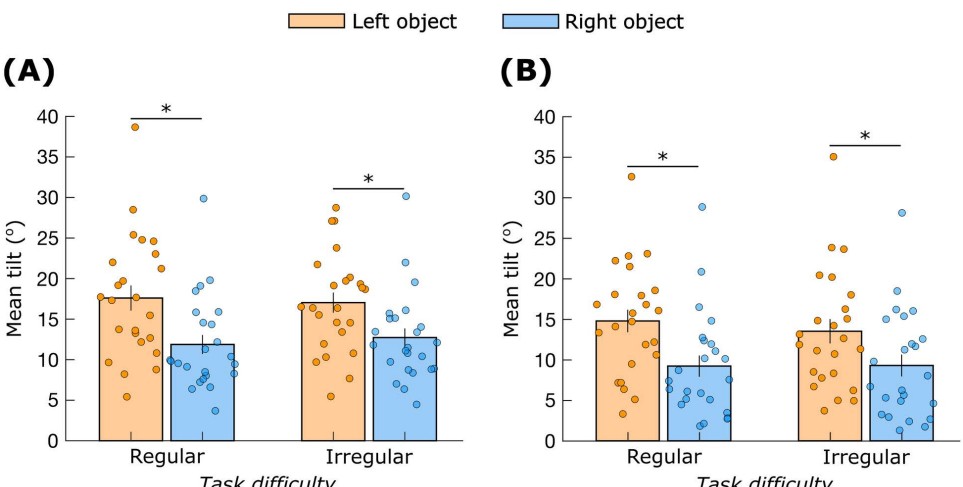

**Fig 7. Mean object tilt (°).** (A) Moving hand, and (B) Static hand. '*' indicates significant differences ($p < 0.05$). Data are mean ± standard error.

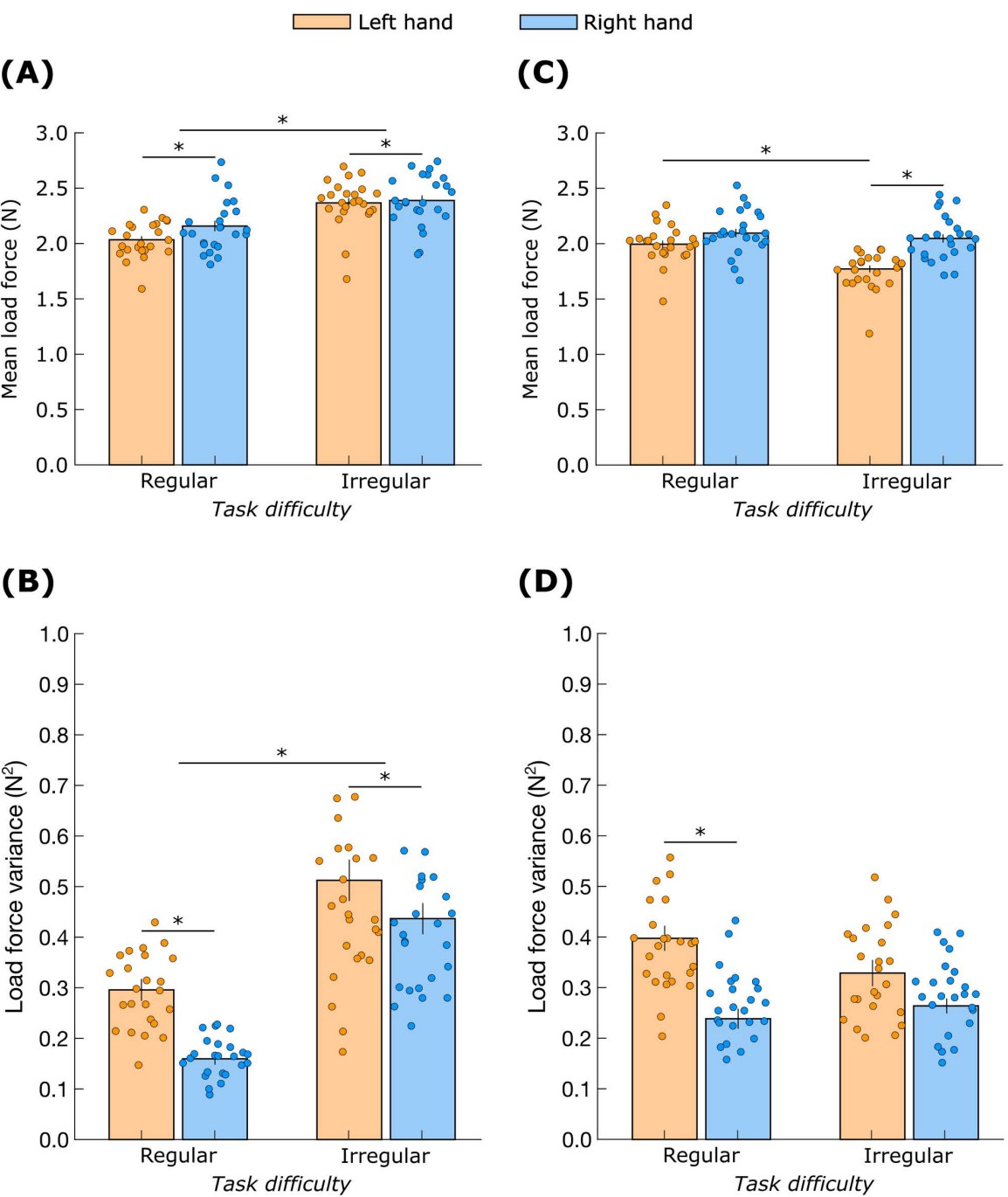

**Fig 8. Load force characteristics.** Mean (A) and variance (B) of the load on the moving object. Mean (C) and variance (D) of the load on the stabilized object. '*' indicates significant differences ($p < 0.05$). Data are mean ± standard error.

The load force variance on the moving object (Fig 8B) was affected by *hand* ($F_{(1,23)} = 19.0$; $p < 0.01$), and *task difficulty* ($F_{(1,23)} = 104.0$; $p < 0.01$). Post-hoc analysis revealed that the load force variance was lower for the right object compared to the left (Cohen's $d = 0.9$) and lower during regular movement compared to irregular movement (Cohen's $d = 2.1$).

The mean load force on the stabilized object (Fig 8C) showed a *hand × task difficulty* interaction ($F_{(1,23)} = 9.8$; $p < 0.01$). Post-hoc analysis revealed that the mean load force was higher for the right object compared to the left but only during irregular movement (Cohen's $d = 1.4$). Furthermore, the mean load force was higher during regular movement compared to irregular movement but only for the left object (Cohen's $d = 1.2$).

The load force variance on the stabilized object (Fig 8D) showed a significant *hand × task difficulty* interaction ($F_{(1,23)} = 6.4$; $p = 0.02$). Post-hoc analysis revealed that the variance was higher for the left object compared to the right but only during the regular movement (Cohen's $d = 1.2$).

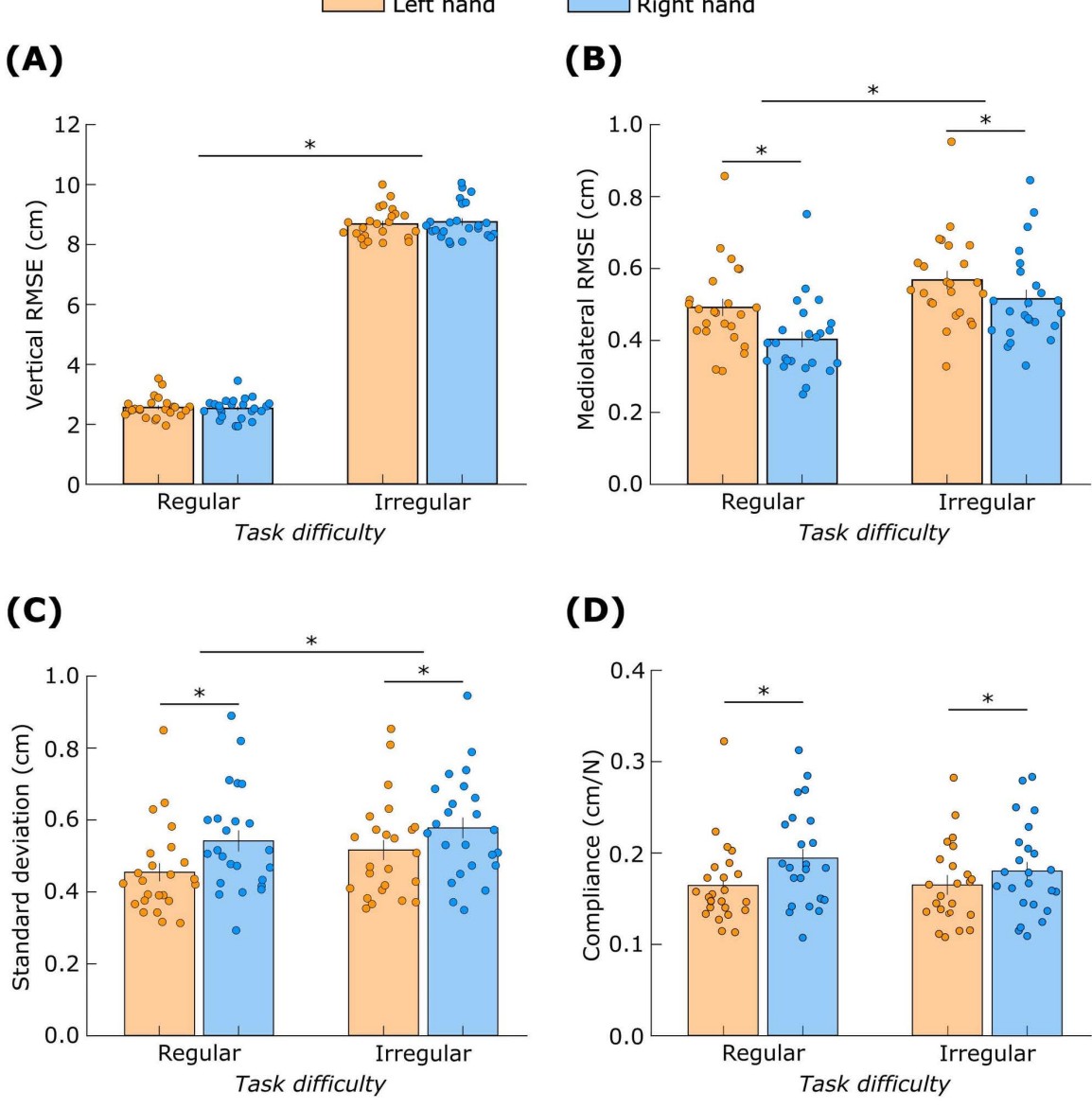

**Fig 9. Task performance measures.** RMSE (A) vertical and (B) mediolateral during object movement. (C) standard deviation and (D) compliance during object stabilization. '*' indicates significant differences ($p < 0.05$). Data are mean ± standard error.

### Task performance measures

The RMSE in the vertical direction for the moving object (Fig 9A) was not affected by *hand* ($F_{(1,23)} = 0.01$, $p = 0.92$) but was affected by *task difficulty* ($F_{(1,23)} = 6128$, $p < 0.01$). Post-hoc analysis revealed that the RMSE was low during regular movement compared to irregular movement (Cohen's $d = 16$).

The RMSE in the mediolateral direction for the moving object (Fig 9B) was affected by *hand* ($F_{(1,23)} = 30.1$, $p < 0.01$) and *task difficulty* ($F_{(1,23)} = 51.8$, $p < 0.01$). The RMSE was low for the object in the right hand compared to the left hand (Cohen's $d = 1.1$) and low for regular movement compared to irregular movement (Cohen's $d = 1.4$).

The standard deviation of the stabilized object's position (Fig 9C) was affected by *hand* ($F_{(1,23)} = 18.0$, $p < 0.01$) and *task difficulty* ($F_{(1,23)} = 8.0$, $p < 0.01$). Post-hoc analysis revealed that standard deviation was low for the object in the left hand compared to the right hand (Cohen's $d = 0.9$) and low for regular movement compared to irregular movement (Cohen's $d = 0.6$).

Finally, the compliance of the stabilized object's position (Fig 9D) was affected by *hand* ($F_{(1,23)} = 10.8$, $p < 0.01$) but not by *task difficulty* ($F_{(1,23)} = 0.9$, $p = 0.35$). Post-hoc analysis revealed that the object stabilized by the left hand was less compliant than the object stabilized by the right hand (Cohen's $d = 0.7$).

### Measures of grip force control

The grip-load correlation coefficient for the moving hand (Fig 10A) showed a significant *hand* × *task difficulty* interaction ($F_{(1,23)} = 6.6$, $p = 0.02$). Post-hoc analysis revealed that the coefficient was higher for the right hand compared to the left hand but only during the regular movement (Cohen's $d = 1.0$). Furthermore, the coefficient was higher for the irregular movement compared to the regular movement but only for the left hand (Cohen's $d = 0.7$).

The non-linear measure Trapping Time for the moving hand (Fig 10B) was affected by *hand* ($F_{(1,23)} = 8.7$, $p < 0.01$) but not by *task difficulty* ($F_{(1,23)} = 1.7$, $p = 0.2$). Post-hot analysis revealed that the trapping time was lower for the right hand than the left hand (Cohen's $d = 0.6$).

The mean grip force on the stabilized object (Fig 10C) was affected by *hand* ($F_{(1,23)} = 34.7$, $p < 0.01$) but not by *task difficulty* ($F_{(1,23)} = 0.55$, $p = 0.45$). Mean grip force was higher for left hand compared to right hand (Cohen's $d = 1.2$).

### Discussion

The primary aim of this study was to determine whether features of complementary dominance (formerly known as dynamic dominance) extend to the movements of hand-held objects and to grip-force control. Our results support the predictions of the complementary dominance theory. The right hand showed superior tracking performance in the mediolateral direction, indicating that the right hand produced straighter movements compared to the left hand, partially supporting hypothesis H1. However, there was no between-hand difference in the tracking performance in the vertical direction. The left hand showed superior stabilizing performance, supporting H2. The grip-load coupling strength during object movement was stronger in the right hand, supporting H3. Finally, the left hand exerted a higher grip force when stabilizing the object, supporting H4. These results demonstrate key differences in the motor performance and grip force control of each hand that reflect their distinct advantages.

The secondary aim of this study was to determine whether task difficulty modulated complementary dominance effects in grip-force control. Task difficulty had a minimal effect on grip force characteristics. The effect of task difficulty on grip-load coupling was not consistent, and the data provided weak support for H5. The linear correlation coefficient between grip and load forces increased with task difficulty but only for the left hand. The non-linear

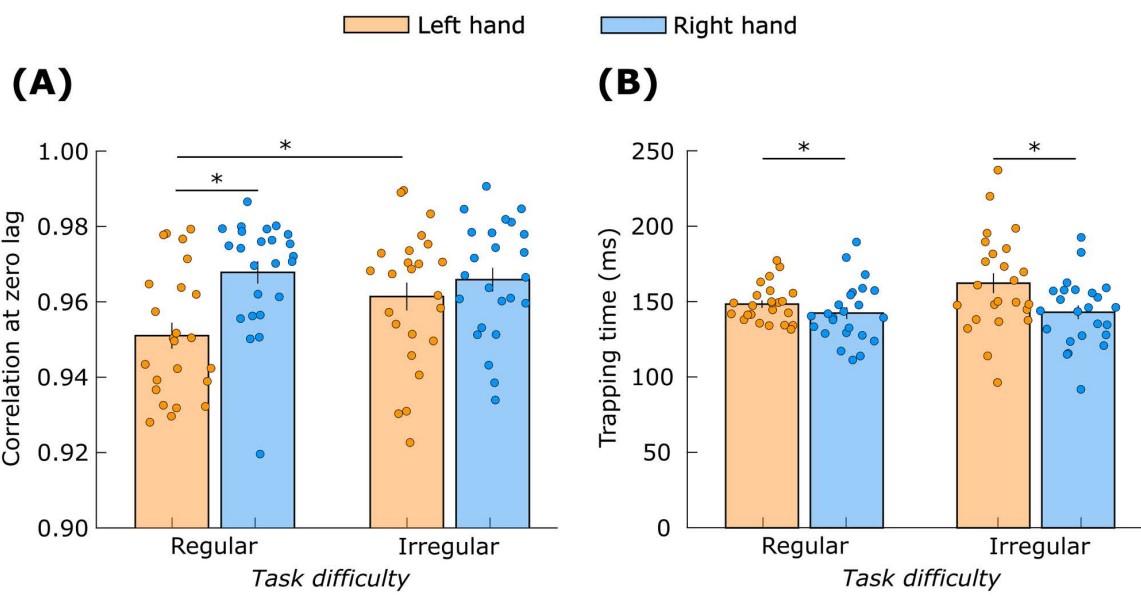

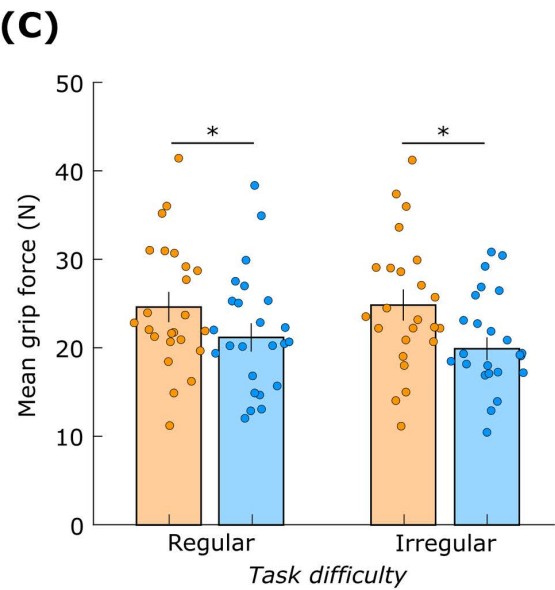

**Fig 10. Grip force characteristics.** Measures quantifying grip-load force coupling during object movement (A) Correlation at zero lag, and (B) trapping time. (C) Mean grip force during object stabilization. '*' indicates significant differences ($p < 0.05$). Data are mean ± standard error.

coupling measure, Trapping Time, was not influenced by task difficulty for either hand. Finally, task difficulty did not affect mean grip force when either hand stabilized the object; there was no support for H6.

## Evidence for complementary dominance in grip force control

The brain's hemispheric division of labor is an ancient feature predating vertebrates [55]. The left hemisphere handles well-established behaviors, while the right hemisphere detects and

responds to unexpected stimuli [56–58]. The division enhances behavioral efficiency. These notions were extended to arm motion control when Sainburg and colleagues developed the complementary dominance theory of handedness [8,41]. They characterized the behavioral consequences of the lateralization in brain function by studying wrist trajectories in healthy individuals and stroke survivors [8,13,41,59–63].

We wanted to determine whether hemispheric specialization was evident in the control of manipulated objects in right-handed individuals. The existing behavioral evidence does not settle this issue. On the one hand, control of arm motion and grip forces is dissociated in some tasks. Humans can alter grip-load coupling without changing arm posture. For example, grip force increases to maintain an object's position in anticipation of an external disturbance [22] and also *before* voluntarily moving the arm to manipulate the object [23]. On the other hand, the tight grip-load coupling while oscillating a hand-held object indicates that the control of grip force is coupled to that of arm motion for that task.

The existing neurophysiological evidence also cannot settle this issue either because the reaching and grasping neural circuits overlap, albeit partially. Some components of the reaching circuit, namely the V6A and the medial intraparietal area [16–18], are distinct from some components of the grasping circuit, namely, the anterior intraparietal area and the ventral premotor cortex [19,21,64,65]. However, other areas, like the dorsal premotor cortex, are included in both circuits [17,19]. This neural architecture may be responsible for the coupling and the decoupling of grip force and arm motion control that is observed for different tasks. Therefore, it was essential to determine whether complementary dominance, which is well established for arm motion control, extends to grip force control. We have provided behavioral evidence supporting its extension, and this is our main contribution to literature.

The stronger grip-load coupling in the right hand (Figs 10A and B) indicates that the left hemisphere's specialization in predictive control influences grip forces. The between-hand difference in this coupling may be attributed to the differences in the efficacy of the predictive control. The existence of grip-load coupling implies that the reaching and grasping neural circuits interact [20]. Therefore, we speculate that the superior predictive control by the left hemisphere may arise from enhanced or better optimized interactions between the circuits in that hemisphere. Further research is necessary to test this hypothesis.

In addition to the stronger grip-load coupling, the object motion was straighter, i.e., with lower medial-lateral deviation, with the right compared to the left hand (Fig 9B). This result matches the straighter wrist paths while reaching with the right arm observed previously [8,9]. The straighter movements suggest better prediction of the arm inter-segmental dynamics and the spring load by the left hemisphere, compatible with earlier work showing straighter paths of the right compared to the left wrist after adapting to predictable external force fields [41].

However, we did not observe between-hand differences in target tracking performance in the vertical direction (Fig 9A). This may be because target tracking requires controlling the object based on visual feedback on the target's position, and hemispheric specialization for visual-spatial processing may have interfered with that for movement control. The right hemisphere is considered superior in visual-spatial processing [66], and it has been suggested that this specialization is evident when experiencing a mismatch between motor intention and visual feedback [67]. We speculate that in the right hemisphere, specialization of visual-feedback-driven tracking compensated for the weaker predictions of arm dynamics and spring loads. Whereas, in the left hemisphere, better predictions of the arm and spring dynamics compensated for the weaker visual-feedback-driven processes. This may have led to similar tracking performance by both hands.

The superior impedance control with the right hemisphere was evident in the lower compliance of the object (Fig 9D), and this was accompanied by higher grip force in the left

hand when stabilizing the object ([Fig 10C]). The higher average grip force would improve the object's impedance by providing better protection against object slip due to unexpected large load force on the object. This view is compatible with the well-known fact that humans produce a grip force that is higher than the minimum required to prevent object slip [26,28,29], and the more recent finding that average grip force is even higher if the load dynamics are less predictable [35].

Furthermore, higher grip force improves the object's impedance by improving the impedance of the hand via co-contraction of the wrist flexors and extensors. Extrinsic hand muscles are involved in producing grip force with a pinch grasp [68,69]. These muscles have their bellies in the forearm, with tendons crossing the wrist [70]. When these muscles contract to create a higher grip force, they also generate a larger wrist flexion moment. To maintain wrist orientation, the wrist extensors must contract, thus generating greater co-contraction. This stiffens the wrist and improves impedance of the hand [29,30].

The lower compliance and higher grip force in the left object and hand ([Figs 9D] and [10C]) reflect complementary hemispheric specialization. The right hemisphere, specialized for impedance control, relies more on this control mode and generates higher grip forces to achieve stability. In contrast, the left hemisphere's specialization and reliance on predictive control leads to lower grip forces and less effective object stabilization.

## Effect of task difficulty on bimanual grip force control

Contrary to our expectation, *task difficulty* had a minimal effect on between-hand differences in grip force characteristics. The hypothesized increase in the grip-load coupling for the irregular motion compared to the regular motion was not consistent, lending only partial support to earlier findings [34]. One reason could be that there was no external load on the object in the previous work [34,71,72], whereas the spring that coupled the two objects induced an external load on the moving object. Furthermore, the load force in the irregular task was less predictable, which may have led to weaker grip-load coupling than expected. We verified that the excursions of the stabilized object that occurred despite the participants' efforts were more complex for the irregular task ([S2 Appendix]). Furthermore, the total load on the moving object was dominated by the spring load: the average spring force (2.95 N) was 17 times larger than the average inertial force (0.17 N), and the variability in the spring load (0.66 N$^2$) was 33 times larger than that in the inertial load (0.02 N$^2$) on the moving object ([S3 Appendix]). Therefore, greater irregularity (or complexity) in the excursions of the stabilized object likely influenced the predictability of the spring and total load on the moving object, which may have negatively impacted the grip-load coupling for the irregular task.

The mean grip force on the stabilized object did not increase during the irregular task compared to the regular task. This result contradicts the finding from a recent study that the mean grip force is three times more sensitive to variability rather than the average value of environmental perturbations [35]. A likely reason could be that the spring load on the stabilized object was not sufficiently unpredictable for the grip force to show significant increase across tasks. The load on the stabilized object was due to voluntary movement of the moving hand exerted via the spring. Furthermore, the spring stiffness was not modulated across trials. Therefore, the motor system could have predicted the spring load arising from the arm motions even during the irregular task. Thus, it is plausible that both feedforward and feedback mechanisms were involved when stabilizing the object against perturbations arising from volitional actions [73], and therefore the effect of load force variability on mean grip force was not observed.

Our plausible explanation for the lack of effect on grip-load coupling is too much unpredictability in the load on the moving object, whereas for the grip force, it is too much

predictability in the load on the stabilized object. This might appear incompatible. However, the unpredictable component of the load on the moving object is thought to arise from *involuntary* movements of the stabilized object, likely due to motor noise and mechanical disturbances in the apparatus. In contrast, the loads on the stabilized object are thought to arise mainly from the *voluntary* movement of the other arm, which the nervous system can predict.

## Contributions to grip force from the central drive

Although our data demonstrates hand-specific differences in object manipulation and grip force control that are consistent with the complementary dominance theory, inter-hemispheric interference/coupling should also be considered while explaining our results [42,43,74,75]. When two hands simultaneously perform asymmetrical tasks, the grip force of both hands tends to be more similar than expected. For example, in a bimanual task in which each hand moves an object of the same weight in one condition, increasing the weight of just one object leads to an increase in the grip force of both hands. This interference occurs because of a common central drive, hypothesized to facilitate bimanual control, that increases the grip force in both hands [42]. It is possible that the grip forces of both hands in our study are affected by the common drive, and therefore, the hand-specific differences in grip force characteristics may not be reflection of just hemispheric specialization.

Nevertheless, hemispheric specialization likely contributes to the patterns in our data. The central drive is stronger and leads to greater equalization of grip forces when the two hands perform asymmetrical tasks, for example, when one hand moves one object while the other hand holds another object static [75], or when both hands move two objects simultaneously but with different loads [42,43]. However, the effects of the central drive are the same when the roles of the hands are switched [42,43]. In contrast, the mean grip force in our study is different when the left versus the right hand stabilized the object (Fig 10C) indicating that both complementary dominance and the common central drive influence grip force control. Future work should quantify the contributions of these two phenomena and their interactions in ecological bimanual tasks.

## Limitations

The first limitation is that our findings are limited to right-handed individuals. Left-handed individuals show more diverse behavior and neurophysiology compared to their right-handed counterparts [76,77]. The laterality indices for left-handed individuals are more variable [78], and left handers demonstrated more balanced hand use indicating a lack of bias toward one hand [77]. This pattern is likely related to left handers exhibiting higher activation of the ipsilateral cortex during unimanual tasks compared to right handers [76]. This suggests that the more symmetrical behavior in left-handers may result from more bilateral hemisphere recruitment, allowing each hand to benefit from specialized functions of each hemisphere [79]. Future research should investigate neural mechanisms of grip force control in left-handed (as well as ambidextrous) individuals to build a comprehensive understanding of how the two hemispheres are specialized for motor control in humans.

Another limitation is that we did not quantify across-hand differences in reflex-mediated grip force responses to changes in the load on the stabilized object. A way to improve object stability, in addition to increasing average grip force, is to increase the gain of the muscular stretch reflex. This has been demonstrated in upper arm studies where motor-cortex-mediated increase in stretch-reflex gain [80,81] enhanced muscular responses to proprioceptive feedback, which led to better resistance to external disturbances [82,83]. Future studies should test hemispheric differences in the modulation of long-latency reflexes and the associated differences in reactive grip forces when the object is unexpectedly perturbed [84,85].

We had argued that the task must be sufficiently challenging to identify differences in control. While our methods successfully demonstrated across-hand behavioral differences in prehensile control, we were not able to identify the impact of modulating task difficulty on the control. This suggests that our manipulation with the regular vs irregular trajectories was not optimal, and there is room for improving this protocol.

Another point worth mentioning is that we consistently observed less object tilt in the right hand compared to the left hand (Fig 7). One could argue that this contradicts H2 and favors superior stabilizing performance with the right hand. We emphasize that we did not provide the participants with feedback on objects' tilt and neither did we explicitly instruct them to maintain vertical orientation of the objects. Therefore, it is difficult to conclude that our object tilt results contradict H2. Nevertheless, this aspect of the control should be explored further.

Finally, we have provided only behavioral evidence in support of complementary dominance in grip force control and object manipulation. Corroborating neurophysiological evidence in the vein of Sainburg and colleagues [14,15] is necessary to conclusively extend the domain of applicability of this theory to prehension.

## Conclusion

This study demonstrates that grip fore control and object manipulation reflect complementary dominance. While simultaneously manipulating two objects coupled by a spring, right-handed individuals demonstrated straighter movements and stronger grip-load coupling while moving an object with the right hand, indicating superior predictive control of the right hand by the left brain hemisphere. In contrast, the object compliance was lower, and the mean grip force on the object was higher when individuals stabilized an object with the left hand, indicating superior impedance control of the left hand by the right brain hemisphere. Task difficulty had a weak effect on grip-load coupling during object movement and no effect on mean grip force during object stabilization, likely due to task design. The load on the moving object and thereby the grip-load coupling was impacted by the complexity of inadvertent excursions of the stabilized object, and external load on the stabilized object was only partially unpredictable since it arose from the voluntary movement of the other hand. Overall, this study provides behavioral evidence highlighting hand-specific specialization of the control of prehensile manipulation. Neurophysiological investigations can now map these differences onto neural substrates and identify the neural control mechanisms.

## Supporting information

**S1 Appendix. Cross-recurrence quantification analysis for grip-load coupling.**
(DOCX)

**S2 Appendix. Complexity of the excursions of the stabilized object.**
(DOCX)

**S3 Appendix. Load force components on moving object.**
(DOCX)

**S4 Appendix. Tracking performance scores.**
(DOCX)

## Author contributions

**Conceptualization:** Anvesh Naik, Satyajit Ambike.

**Data curation:** Anvesh Naik.

**Formal analysis:** Anvesh Naik.

**Investigation:** Anvesh Naik, Satyajit Ambike.

**Methodology:** Anvesh Naik, Satyajit Ambike.

**Project administration:** Satyajit Ambike.

**Resources:** Satyajit Ambike.

**Software:** Anvesh Naik.

**Supervision:** Satyajit Ambike.

**Validation:** Anvesh Naik, Satyajit Ambike.

**Visualization:** Anvesh Naik.

**Writing – original draft:** Anvesh Naik, Satyajit Ambike.

**Writing – review & editing:** Anvesh Naik, Satyajit Ambike.

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
