## [Decision Letter · Decision Letter 0]

8 Nov 2024

PONE-D-24-37850Handy divisions: Hand-specific specialization of prehensile control in bimanual tasks

PLOS ONE

Dear Dr. Ambike,

Thank you for submitting your manuscript to PLOS ONE. After careful consideration, we feel that it has merit but does not fully meet PLOS ONE’s publication criteria as it currently stands. Therefore, we invite you to submit a revised version of the manuscript that addresses the points raised during the review process.

Please address with particular emphasis the concerns raised by both reviewers regarding the interpretation of results, specifically focusing on:

The speculations of neurophysiologic mechanisms underlying the observed behavioral phenomena.

The distinction between hemispheric dominance and interhemispheric differences, ensuring conceptual clarity.

Additionally, please attend to the readability issues highlighted by Reviewer 1 

We look forward to receiving your revised manuscript.

Kind regards,

Maria Nazarova

Academic Editor

PLOS ONE

Journal requirements:    When submitting your revision, we need you to address these additional requirements. 1. Please ensure that your manuscript meets PLOS ONE's style requirements, including those for file naming. The PLOS ONE style templates can be found at https://journals.plos.org/plosone/s/file?id=wjVg/PLOSOne_formatting_sample_main_body.pdf and https://journals.plos.org/plosone/s/file?id=ba62/PLOSOne_formatting_sample_title_authors_affiliations.pdf 2. Thank you for stating the following financial disclosure:  [Anvesh Naik was funded by The Ross Fellowship awarded by Purdue University.].  Please state what role the funders took in the study.  If the funders had no role, please state: ""The funders had no role in study design, data collection and analysis, decision to publish, or preparation of the manuscript."" If this statement is not correct you must amend it as needed. Please include this amended Role of Funder statement in your cover letter; we will change the online submission form on your behalf. 3. We notice that your supplementary figures are uploaded with the file type 'Figure'. Please amend the file type to 'Supporting Information'. Please ensure that each Supporting Information file has a legend listed in the manuscript after the references list.

Reviewers' comments:

Reviewer's Responses to Questions

**Comments to the Author**

1. Is the manuscript technically sound, and do the data support the conclusions?

Reviewer #1: Partly

Reviewer #2: Yes

2. Has the statistical analysis been performed appropriately and rigorously? 

Reviewer #1: I Don't Know

Reviewer #2: Yes

3. Have the authors made all data underlying the findings in their manuscript fully available?

Reviewer #1: Yes

Reviewer #2: Yes

4. Is the manuscript presented in an intelligible fashion and written in standard English?

Reviewer #1: Yes

Reviewer #2: Yes

5. Review Comments to the Author

Reviewer #1: # General comments

## Is the manuscript clearly written?

Overall, the manuscript is clearly written. It is easy to read for the experts, who are familiar with the terms and concepts used in the field. However for the general scientific audience with no background in kinesiology (like myself) understanding the manuscript requires a fair amount of googling to learn the concepts that the authors use (e.g. "impedance", "load", "coupling", etc.). I find it perfectly normal that understanding a scientific paper requires a significant amount of expertise in the paper's domain; nevertheless, I think the authors can greatly increase the paper's appeal to a broader scientific audience by providing a brief introduction to the terms unfamiliar to the readers from outside the field. For example, a table with a few sentences briefly describing each term may be very useful.

## Is the experiment described in the manuscript technically sound?

The manuscript describes a rigorous and technically sound experiment for measuring several kinematic and kinetic variables (grip force, grip force-load force coupling, and various kinematic variables) and studying the influence of different experimental conditions on these variables. The statistical methods used by the authors generally seem reasonable to me, however I lack the expertise to make any definite comments on them.

## Do the results presented in the manuscript support the conclusions?

This is my biggest beef with this manuscript. Whereas the authors do a good job of measuring the biomechanical variables of interest under several experimental conditions, their interpretation of the results of these measurements as evidence for the dynamic dominance theory is not convincing enough, at least for a reader like myself, who is not an expert in the field. In particular:

* The authors interpret the higher grip force in the stabilizing hand as the evidence for better impedance control. While this may be true, it's not immediately obvious. From a purely mechanical perspective, higher grip force itself does not impact the stability of instrumented object (as long as the force is sufficient to prevent the object from slipping from the hand). What matters for the object's stability is the efficient use of the corrective forces that the hand applies in response to the unexpected displacement of the object. I would expect a more efficient use of these corrective forces -- rather than the average grip force -- to be a sign of a better impedance control by a particular hemisphere.

It's worth noting that the authors suggest the wrist stiffness induced by the muscle co-contraction as a partial explanation for the link between the the higher grip force and better impedance control. It is not obvious how this explanation supports the conclusion that that higher grip force indicates better hemispheric specialization for impedance control. If anything, one might argue that it actually __weakens__ the authors' conclusions by providing an alternative explanation.

* The authors assume that in their setup the participant will be able to track the moving target better when the tracking hand is controlled by a hemisphere specializing for predictive control; similarly they assume that the subject will be able to stabilize the instrumented object better when the stabilizing hand is controlled by a hemisphere specializing in reactive control. It's worth noting that other publications (e.g. reference 12) rely on similar assumptions. Nevertheless it's not clear to me why this assumption should hold. Two objects connected by a spring is a very simple mechanical system; to me it looks quite possible that the participants can learn to model it using the same predictive control mechanisms they use for modelling their hands. If this is the case, both tasks (tracking the target and stabilizing the object) require equal amount of predictive and reactive control; one cannot expect that participants will perform one task better when using a hemisphere specialized for predictive control and the other -- when using a hemisphere specialized in reactive control. The authors are clearly aware of this problem and discuss it in the "Discussion" section, e.g. lines 460 -- 465. Yet they fail to convincingly explain why their conclusions are not invalidated by such a possibility.

# Specific comments

* The code and data for the manuscript are available from doi:10.4231/GB7J-RA71. The zip archive downloaded through this DOI seems to contain both the data recorded by the authors and the code they used in there analysis. However, I failed to find within the archive any meaningful documentation explaining how the data is organized or how to use the code to reproduce their results. The lack of such documentation greatly reduces the usefulness of the code and data provided, and I would recommend the authors to provide a comprehensive documentation of their code and data if they have enough resources. Nevertheless, considering relative costs and benefits of properly documenting the code, I think this should be taken as a recommendation and not a mandatory requirement for publication.

* I recommend replacing the term "global dominance theory" with the "global hemispheric dominance theory" throughout the text to make the terminology more consistent with the literature.

* I suggest replacing the heading "Equipment" (line 98) with "Instrumentation" or something similar.

* For the difficult condition it seems that the same sequence of brownian motion (mentioned in the lines 154 -- 155) was used within a single test subject (otherwise it would be impossible to compute meaningful std). Was the same sequence used for all the subjects? I recommend explaining it more clearly.

* Caption for the Fig.5: I recommend explicitly emphasizing the fact the the plots use different axis for the grip and load forces.

* Lines 281 and further down the text: I find the notation F(1,23) confusing. Can it be a copy-paste error? I suggest either correcting it or explaining it more clearly.

* Line 293 and further down the text: I suggest adding the degree sign whenever reporting the angles.

* I think that moving the technical details of CRQA to the appendix will make the manuscript easier to read.

Reviewer #2: Thank you for the opportunity to review the manuscript by Naik and Ambike entitled “Handy divisions: Hand-specific specialization of prehensile control in bimanual tasks”.

The aim of this study was to investigate whether object manipulation supports the dynamic dominance theory. This theory suggests that the left hemisphere specializes in predictive control, making the right hand more adept at coordinated movement, while the right hemisphere specializes in impedance control, making the left hand better at stabilization. Previous studies using wrist kinematics have supported this theory and this work seeks to utilize object orientation and grip force measures.

The authors designed an instrumented object for each hand, linked by a spring, that measured pinch force and position. The position of each object was represented by cursors on a screen. Participants were instructed to maintain one hand’s position within a stationary target while using the other hand to track a moving target. The moving target followed either a predictable or irregular pattern, and participants completed trials with either hand serving as the stabilizer or the tracker.

Performance measures included basic performance metrics, task-specific measures, and grip force control measures. The findings showed that the right hand excelled in tracking performance, while the left hand was better at stabilization. They also observed stronger grip-load coupling—a marker of predictive control—in the right hand during object movement, while mean grip force—a marker of impedance control—was stronger in the left hand during stabilization. Task difficulty did not significantly impact force characteristics.

The authors are to be commended on the overall rationale, design and presentation of their study. The manuscript was well written, and the figures were well presented.

Main Comments

1. My primary concern with this study is the absence of left-hand dominant participants and the lack of neuroimaging data, both of which impact the strength of conclusions about neurophysiology and hemispheric specialization. While the authors do connect their findings to previous studies, without data from left-hand dominant participants or neuroimaging, it’s challenging to make definitive connections between observed behaviors and neurophysiological mechanisms.

a. Including the discussion of limitations around the exclusion of left-hand dominant individuals is valuable. However, the authors also reference prior work showing distinct neural organization for left-hand dominant individuals, which seems to contradict their generalized claims about hemispheric specialization earlier in the paper. This discrepancy highlights the need for a more cautious interpretation of hemispheric specialization, as it appears to depend on hand dominance.

b. (Lines 394-399) Brain structures are listed to emphasize the fact that arm motion and digit forces/grasping control is controlled by distinct structures. However, dorsal premotor cortex is included for both. I would suggest emphasizing the distinct neural pathways in the anterior and medial intraparietal areas.

i. Lines 403-404 then go on to say that reaching and grasping neural circuits interact. I would suggest clarifying language because the dissociation discussed in one paragraph seems to contradict the interaction in the next paragraph.

2. Figure 7 indicates that, regardless of task difficulty or whether the task involves tracking or stabilizing, the right hand consistently shows reduced mean tilt. Since mean tilt is used here as a measure of stabilization, this finding seems counter to the hypothesis that the right hemisphere, and thus the left hand, would specialize in stabilization. I suggest further elaboration on this unexpected result and its implications, as it does not appear to support the main hypothesis.

Minor Comments

1. Vague language used in introduction: “dominant (right) hand is better…” (line 4), “superiority of the left hemisphere…” (line 6)

2. Elaborate on inclusion/exclusion criteria to indicate whether individuals with any motor or sensory impairments were excluded

3. Add the icons used in figure 5 to indicate which object moved and which remained stationary to figure 6

4. Figure 9D has smaller points than the other subplots

Overall, I think this is a well written manuscript and highlights important results. The minor comments should be resolved, and the neurophysiological conclusions drawn from behavioral results and prior literature without imaging data related to the study participants should be minimized.

6. PLOS authors have the option to publish the peer review history of their article (what does this mean? ). If published, this will include your full peer review and any attached files.

**Do you want your identity to be public for this peer review?** For information about this choice, including consent withdrawal, please see our Privacy Policy .

Reviewer #1: No

Reviewer #2: No

---

## [Author Response · Author response to Decision Letter 0]

16 Jan 2025

We have responded to all editor and reviewer comments. These responses are contained in the Response to reviewers document uploaded earlier.

---

## [Decision Letter · Decision Letter 1]

11 Mar 2025

Handy divisions: Hand-specific specialization of prehensile control in bimanual tasks

PONE-D-24-37850R1

Dear Dr. Ambike,

We’re pleased to inform you that your manuscript has been judged scientifically suitable for publication and will be formally accepted for publication once it meets all outstanding technical requirements.

Kind regards,

Maria Nazarova

Academic Editor

PLOS ONE

Additional Editor Comments (optional):

Please consider changing the letter "I" to "E" inthe word "complementary" throughout the manuscript.

Reviewers' comments:

Reviewer's Responses to Questions

**Comments to the Author**

1. If the authors have adequately addressed your comments raised in a previous round of review and you feel that this manuscript is now acceptable for publication, you may indicate that here to bypass the “Comments to the Author” section, enter your conflict of interest statement in the “Confidential to Editor” section, and submit your "Accept" recommendation.

Reviewer #1: All comments have been addressed

Reviewer #2: All comments have been addressed

2. Is the manuscript technically sound, and do the data support the conclusions?

Reviewer #1: Yes

Reviewer #2: Yes

3. Has the statistical analysis been performed appropriately and rigorously? 

Reviewer #1: I Don't Know

Reviewer #2: Yes

4. Have the authors made all data underlying the findings in their manuscript fully available?

Reviewer #1: Yes

Reviewer #2: Yes

5. Is the manuscript presented in an intelligible fashion and written in standard English?

Reviewer #1: Yes

Reviewer #2: Yes

6. Review Comments to the Author

Reviewer #1: I suggest replacing "complimentary dominance" with "complementary dominance" throughout the manuscript.

Reviewer #2: Thank you for addressing all the comments thoroughly. This manuscript has been significantly strengthened - particularly in the discussion with interpretation of neurophysiological-behavioral connections and highlighting limitations. The addition of the key terms table in the introduction is also very helpful to orient the reader to all the terms used.

7. PLOS authors have the option to publish the peer review history of their article (what does this mean? ). If published, this will include your full peer review and any attached files.

**Do you want your identity to be public for this peer review?** For information about this choice, including consent withdrawal, please see our Privacy Policy .

Reviewer #1: **Yes: ** Andrey Zhdanov, PhD, https://orcid.org/0000-0002-3379-0251

Reviewer #2: No

---

## [Editor Report · Acceptance letter]

PONE-D-24-37850R1

PLOS ONE

Dear Dr. Ambike,

I'm pleased to inform you that your manuscript has been deemed suitable for publication in PLOS ONE. Congratulations! Your manuscript is now being handed over to our production team.

Kind regards,

on behalf of

Dr. Maria Nazarova

Academic Editor

PLOS ONE